# Fabrication of Nanofibers Based on Hydroxypropyl Starch/Polyurethane Loaded with the Biosynthesized Silver Nanoparticles for the Treatment of Pathogenic Microbes in Wounds

**DOI:** 10.3390/polym14020318

**Published:** 2022-01-13

**Authors:** Mohamed E. El-Hefnawy, Sultan Alhayyani, Mohsen M. El-Sherbiny, Mohamed I. Sakran, Mohamed H. El-Newehy

**Affiliations:** 1Department of Chemistry, Rabigh College of Sciences and Arts, King Abdulaziz University, Jeddah 21589, Saudi Arabia; salhayyani@kau.edu.sa; 2Marine Biology Department, Faculty of Marine Sciences, King Abdulaziz University, Jeddah 21589, Saudi Arabia; ooomar@kau.edu.sa; 3Biochemistry Department, Faculty of Science, University of Tabuk, Tabuk 47731, Saudi Arabia; msakran@ut.edu.sa; 4Department of Chemistry, Faculty of Science, Tanta University, Tanta 31527, Egypt; 5Department of Chemistry, College of Science, King Saud University, Riyadh 11451, Saudi Arabia

**Keywords:** hydroxypropyl starch nanofibers, polyurethane, wound dressing, silver nanoparticles, toxicological performance assay

## Abstract

Fabrication of electrospun nanofibers based on the blending of modified natural polymer, hydroxyl propyl starch (HPS) as one of the most renewable resources, with synthetic polymers, such as polyurethane (PU) is of great potential for biomedical applications. The as-prepared nanofibers were used as antimicrobial sheets via blending with biosynthesized silver nanoparticles (AgNPs), which were prepared in a safe way with low cost using the extract of Nerium oleander leaves, which acted as a reducing and stabilizing agent as well. The biosynthesized AgNPs were fully characterized by various techniques (UV-vis, TEM, DLS, zeta potential and XRD). The obtained results from UV-vis depicted that the AgNPs appeared at a wavelength equal to 404 nm affirming the preparation of AgNPs when compared with the wavelength of extract (there are no observable peaks). The average particle size of the fabricated AgNPs that mediated with HPS exhibited a very small size (less than 5 nm) with excellent stability (more than −30 mv). In addition, the fabricated nanofibers were also fully characterized and the obtained data proved that the diameter of nanofibers was enlarged with increasing the concentration of AgNPs. Additionally, the findings illustrated that the pore sizes of electrospun sheets were in the range of 75 to 350 nm. The obtained results proved that the presence of HPS displayed a vital role in decreasing the contact angle of PU nanofibers and thus, increased the hydrophilicity of the net nanofibers. It is worthy to mention that the prepared nanofibers incorporated with AgNPs exhibited incredible antimicrobial activity against pathogenic microbes that actually presented in human wounds. Moreover, *P.*
*aeruginosa* was the most sensitive species to the fabricated nanofibers compared to other tested ones. The minimal inhibitory concentrations (MICs) values of AgNPs-3@NFs against *P.*
*aeruginosa*, and *E.*
*faecalis*, were 250 and 500 mg/L within 15 min, respectively.

## 1. Introduction

Nanomaterials including inorganic nanoparticles, organic materials, or nanocomposites, could be presented as bio-nanomaterials, colloidal particles, copolymers, and gels. Recently, nanofibers were used to integrate inorganic nanoparticles and nanocomposites into scaffolds for wound healing applications [1,2,3]. On the other hand, silver nanoparticles (AgNPs) as inorganic nanoparticles have a numerous range of applications in the biomedical sector due to their several benefits including a straightforward preparation method, a facile approach to control its morphology, and a high specific surface area to volume ratio [4,5,6,7,8,9,10]. Over the last decades, many published documents have been focused on the antimicrobial efficacy of AgNPs against microbial species [11,12,13]. The performance of AgNPs towards microorganisms was determined by essential aspects, such as the technique of production and the type of application. Generally, AgNPs have exposed superior antibacterial action against Gram-positive and Gram-negative bacteria [14]. Moreover, AgNPs as antimicrobial agents have minimal toxicity to human cells, making them suitable for therapeutic applications [15] and can be extensively employed for the treatment of wounds and burns via prohibiting pathogens from entering them [16].

Biosynthesis of AgNPs can be attained via using plant extract instead of the utilization of chemical compounds, which can maximize the demand for green chemistry in our daily life [17,18]. *Nerium oleander* is a member of the dogbane family (Apocynaceae) and it is considered as the only species of the genus *Nerium*. The name Oleander comes from its resemblance to olive Olea [19] and it is widely cultivated and no precise region of origin has been identified but southwest Asia has been suggested. The extract of *Nerium oleander* leaves is used for the biosynthesis of silver nanoparticles [20]. The motivation for using this plant can be attributed to its ornamental usage and heavy distribution everywhere, which makes it a very cheap source to be used for the preparation of nanoparticles, particularly, AgNPs.

In order to prepare an efficient antimicrobial nanofibers sheet, nanofibers loaded with AgNPs can be fabricated using the electrospinning technique, which is known as a powerful and cost-effective tool for producing multifunctional nanofibers. Electrospun nanofibers were designed to present distinctive properties, such as a high surface area to volume ratio and nanoscale pore diameters [21,22,23]. Although most polysaccharides are of potential relevance for electrospinning and have been proven to be beneficial in a variety of medicinal purposes, electrospinning exhibited certain limitations [21,24,25,26,27], such as poor mechanical properties, high surface tension, and low solubility in organic solvents. To avoid such issues concerning the electrospinning of polysaccharides, efficient nanofibers can be obtained from modified natural polymers blended with synthetic polymers [28]. The modified natural polymers will have the advantage of solubility in water with no need for organic solvents [29]. In addition, synthetic polymers can be used to enhance the mechanical properties of natural fibers [30].

Modified starch (hydroxypropyl starch, HPS) is one of the most renewable resources that can be obtained by the propylation reaction of starch [13,31,32,33]. HPS has the advantages of nontoxicity and biocompatibility [34]. On the other hand, polyurethane (PU), is employed in biomedical applications due to its excellent mechanical characteristics, and processing adaptability [35,36]. There were also several studies that suggested the utilization of PU as a good candidate for the fabrication of electrospun nanofiber sheets.

Herein, the designed research work aimed to fabricate wound dressing from electrospun nanofibers based on blending the modified natural polymer (hydroxypropyl starch, HPS) and synthetic biopolymers (polyurethane, PU) with the biosynthesized AgNPs. The extract of *Nerium oleander* leaves (In Egypt, *Nerium oleander* leaves are naturally presented in abundance on the sides of highways and deserts) was used for the preparation of AgNPs to avoid the utilization of hazardous and expensive chemical compounds for such preparation. The as-prepared extract, AgNPs and nanofibers loaded with AgNPs were characterized in terms of particle shape, hydrodynamic size, zeta potential, morphological structure, roughness contact angle using advanced tools, such as UV-Vis, TEM, DLS, and SEM techniques. The work was extended to extensively examine the antimicrobial features of the as-fabricated nanofibers.

## 2. Materials and Methods

### 2.1. Materials

Hydroxypropyl starch (HPS; Formula weight 666.57768, CAS No. 9049-76-7) was purchased from Shanghai Time Chemicals CO., Ltd. Changhai (China). Polyurethane (PU; *M_w_* = 80 kg mol^−1^) was bought from Sigma-Aldrich Co. Saint Louis MO 63,103 (USA). Silver nitrate (AgNO_3_) was purchased from Sigma Aldrich Co, Berlin, Germany. *N*,*N*-Dimethylforamide (DMF) was supplied by Sigma Aldrich Co., Czech republic, Prague. Dimethyl sulfoxide (DMSO) was provided by Merck KGaA, Berlin, Germany. Glutaraldehyde solution was purchased from WINLAB Co., Kolkata-711102, West Bengal, India. Green leaves of the street ornamental plant *Nerium oleander* (Family: Apocynaceae) and commonly named “Dafla” were collected during September 2021 from Dekernis city, Dakahlia governorate, Egypt.

### 2.2. Biosynthesis of AgNPs

Silver nanoparticles have been biosynthesized by the method previously described by Subbaiya et al., [19] with some modifications. Briefly, the fresh leaves of Nerium oleander (ornamental plant) were collected from plants grown in the streets of Mansoura city, Egypt during September 2021. For preparing the plant extract, Leaves were washed with tap water to remove dirt and dust followed by wiping with filter paper to remove any residual water drops. The plant extract was prepared by mixing 20 g of green leaves with 500 mL of boiling distilled water for 30 min. the produced water extract was filtered through Whatman No. 40 filter paper to get rid of any particulate materials. For the biosynthesis of silver nanoparticles, 25 mL of 2 mM aqueous solution of silver nitrate was added to 50 mL of the extract with continuous stirring at 25 °C for 24 h. During stirring, it was noted that the colorless solution was turned into a brownish red color signifying the formation of AgNPs.

### 2.3. Electrospinning of HPS/PU Nanofibers Blend Lioaded with AgNPs (AgNPs@NFs)

Polymer solutions of HPS (15 wt%) and PU (12 wt%) were prepared separately by dissolving the corresponding weight of polymers in DMSO and DMF, respectively, with vigorous stirring at 70 °C for 2 h. HPS and PU solutions were blended using vigorous stirring and heating with different ratios as shown in Table 1. The obtained admixture was ultrasonicated for 60 min, followed by dropwise addition of different volumes of AgNPs (Table 1) to the polymer admixture prior to the electrospinning process.

The electrospinning process was performed at 35 °C with a voltage of 30 kV, a tip-to-collector distance (TCD) of 10 cm, with a flow rate of 0.1 mL/h (Table 1). The fabricated nanofibrous sheets were treated with glutaraldehyde (35%, 10% *v*/*v*, 1 mL) and were incubated for 12 h followed by incubation in a glycine solution (0.1 M; 2 mL) for 12 h in order to eliminate any residual or unreacted glutaraldehyde. Finally, the nanofibrous samples were rinsed thrice with water and phosphate buffer solution (PBS), respectively to eliminate the excess glycine or glutaraldehyde and were kept at 25 °C for drying.

### 2.4. Physicochemical Characterization of the Extract, AgNPs and Nanofibers

Kumar et al. [37], used the Folin-reagent Ciocalteu’s technique to determine the total phenolic content of Nerium oleander extract. The reaction mix included 100 L of plant extract at 200 g/mL, 500 L of Folin-reagent, Ciocalteu’s and 1.5 L of sodium carbonate (20%). Using distilled water, the mixture was mixed and produced up to 10 mL. After allowing the mixture to sit for two hours, the absorbance was measured at 765 nm. The standard was gallic acid. All of the tests were performed in three different ways. The total phenolic content of the extract was measured in milligrams of gallic acid equivalent (GAE) per of extract.

GC-MS was used to investigate the composition of the extract according to the procedure reported by Madkour et al. [37], using Thermo Scientific, Trace-GC, Ultra/ISQ Single Quadrupole MS, TG-5MS fused silica capillary column (30 m, 0.251 mm, 0.1 mm film thickness).

UV–visible spectrophotometry was used to confirm the production of AgNPs using a UV1601 instrument (Shimadzu, Japan) over the wavelength range of 300 to 700 nm.

A Phillips PW 1830 apparatus operating at a voltage of 40 kV with Cu *K_a_* radiation was used to perform X-ray diffraction (XRD) measurements of the produced AgNPs using the green technique on drop-coated films of the corresponding solutions onto glass substrates. The powdered sample was prepared as follows: 100 mL of biosynthesized AgNPs were centrifuged at 10k rpm for 6 h, and the resultant powder was dried at 70 °C and used for XRD measurements. The sample was analyzed with 2*θ* between 5–80° using Cu *K_α_* radiation generated at 35 mA, 24 kV and 5 °/min. The average crystallite diameter was calculated using the Debye–Scherrer equation: D=Kλ/(βcosθ), where D is a symbol for the particle size (Å), Scherrer’s constant *K* is a shape-dependent constant (1.05). The full peak width is *β* (rad), λ is the radiation’s wavelength, (*θ* rad) is the diffraction angle.

TEM studies were conducted by means of TEM (JEOL TEM 2100) at an accelerating voltage of 100 kV at 25 °C to obtain information about the particle shape of AgNPs. The solution sample was sonicated for 15 min to affirm the good dispersion. Then, a drop of the sonicated sample was placed onto a copper coated grid and left to dry before evaluation.

A particle size analyzer was used to assess the particle size (Malvern Zetasizer nanosizer). Laser Doppler electrophoresis was used to evaluate the zeta potential values. The evaluations were conducted out at 298 K using a Zeta Sizer Non series-ZS (Malvern Instrument, Malvern, UK).

Field emission scanning electron microscopy (FESEM, QUANTA-FEG250, The Netherlands) was used to examine the morphology and diameter of the fabricated electrospun nanofibers. A sputter coater was used to apply a thin layer of gold to the samples. EDX analysis of nanofibers loaded with AgNPs was carried out using FESEM equipped with an EDAX attachment

The surface roughness of the prepared n’nofibrous mats were studied using Gwyddion 2.45 software.

The static contact angle was measured with a portable instrument (Kruss, Hamburg, Germany). For evaluating the contact angle, a swing pump sprays a droplet of deionized water onto a clean surface of the nanofibers cheat in a closed cage.

The infrared spectra were recorde” by utilizing a Burker LUMOS-FTIR Microscope (Bruker Optik GmbH, Ettlingen, Germany) equipped with an ATR reflection module and a diamond crystal, with a single 45-degree reflection and OPUS 8 software for spectral analysis. HPS and PU powder samples, as well as nanofiber mats of both polymers loaded with AgNPs, were examined in the 400–4000 cm^−1^ range. A total of 64 scans at a resolution of 3 cm^−1^ were used to acquire all of the spectra. The IR spectroscopy for all samples was performed at 25 °C.

A Mettler Toledo STARe System was used to obtain the TGA profiles for the selected nanofibers samples. The processes were conducted by taking around 5 mg of each nanofibers samples with a heating rate of 10 °C/min and a temperature range of 0–500 °C under nitrogen atmosphere with taking in mind that the gas flow is 85 mL/min).

### 2.5. Biological Characterization of the Prepared Nanofibers

#### 2.5.1. Tested Microorganisms

The potential antimicrobial activity of the fabricated nanofibers (AgNPs-0@NFs, AgNPs-1@NFs, AgNPs-2@NFs, and AgNPs-3@NFs) was conducted using the agar disc diffusion assay against *Pseudomonas aeruginosa*, *Enterococcus faecalis*, *Candida albicans*, and *Aspergillus niger*, which are among the four worrisome pathogens that pose a significant public health threat.

#### 2.5.2. Antimicrobial Test

Fresh culture of the tested pathogens was obtained after incubation overnight and was fixed to 0.5 McFarland values before being diluted ten times (1:10) to provide 2.3 × 10^6^ CFU/mL. A hundred µL of the prepared microbial suspension of particular pathogens was distributed carefully over the top layer of Muller Hinton agar (MHA). After that, pieces (1 cm × 1 cm) of each of the studied nanofibers were placed carefully on the inoculated MHA medium. Each piece of nanofiber was individually placed at an equal distance to avoid interference on the inoculated medium.

The drug/antibiotic (ciprofloxacin 30 μg) was employed as a positive control, while the negative control was sterile distilled water.

All inoculated dishes were preserved at ambient temperature for 1 h to allow the ingredients for pre-diffusion before incubation aerobically at 37 °C overnight. The formed zone of inhibitions (ZOI) against the chosen pathogens were measured using a calibrated Vernier Caliper in millimeters (mm), documented values. All trials were conducted in triplicates [38].

The Macro-dilution technique was performed to determine the MIC values of the studied nanofibers. Various masses of studied nanofibers (100, 250, 500 mg) was tested against the above-mentioned human-associated pathogens. The beginning cell populations were estimated using the log phase of each selected microbes (log_10_ 6.41 CFU/mL). The remaining microbial cells were measured before and after exposure to the studied nanofibers at multiple time intervals using bacterial cell viability (0, 5, 15, 30 min). Colonies that had already developed were recorded using a colony counting device after all Petri plates were subsequently deposited in the incubator and providing the appropriate conditions for each microbial species. All experiments were conducted thrice in alternating time scales, with the final result obtained at an average of three experiments.

Growth rate and protein released experiments were applied to investigate alterations in the evolution of the physiological behavior of each recognized microbial pathogen. Three tubes holding 25 mL of sterile nutrient broth injected with 100 mL of each target microbe were injected with the selected lethal dose of each NFs. One tube was utilized as a negative control without nanomaterial. All tubes were maintained in a shaking incubator at 37 °C, and 200 rpm for 24 h, with a 1 mL sample collected every 2 h (*n* = 12 readings). The spectrophotometer measured the absorption spectrum to estimate the growth of pathogens. The quantity of intracellular protein free from damaged cells was quantified using Coomassie Blue assay under the same bacterial growth curve procedures as above [39].

The inactivation speed (*k1*) was assumed for all recognized pathogens using pseudo-first-order (*Nt*) kinetic modeling to determine how quickly the selected dose of nanofiber is potent in eliminating and destroying microbial cells.

#### 2.5.3. Toxicological Screening

The toxic effects of the nanostructured materials under study were measured using a Microtox 500 analyzer (Modern Water Inc., New Castle, DE 19720, USA). The cytotoxic effects analysis was performed by reducing the luminescent brightness levels of the bacterial suspension of marine *Vibrio fischeri* bacteria. The recommended dose of each AgNPs@NFs was implanted into a cuvette having 10 µL of the luminescent bacterial suspension. Decreasing luminescent light means that the bacterial mortality was verified in two distinct assays. Dose-response measures are applied to estimate the dose of an effective dose that results in a 50% reduction in bioluminescence or EC_50%_ [40].

### 2.6. Statistical Analyses

GraphPad Prism version 5.0 (USA) was applied to conduct the statistical study. The level of significance (a probability of *p* < 0.05) between the doses of tested nanomaterial and viable cells of targeted microbes was calculated by applying two-way analysis (ANOVA) of variance.

## 3. Results and Discussion

The current study aimed to achieve three main objectives. The first objective is to prepare AgNPs using green chemistry with the aid of plant extract. This plant extract comprises many compounds, aldehydes and alcohols, with reducing groups, such as 5-octadecenal, 1-tridecanol, 1-hexacosanol and 1-docosanol, which can reduce silver ions (Ag^+^) to silver nanoparticles (AgNPs). The extract bearing also hydroxyl groups that can act as stabilizing agents for the produced AgNPs. The second objective is to incorporate AgNPs into electrospun nanofibers of the HPS/PU blend using the electrospinning technique. The final objective is to evaluate the as-prepared electrospun nanofibers as wound dressing sheets. These objectives are illustrated in Figure 1.

### 3.1. GC-MS Investigation of the Plant Extract

Firstly, via determining the total phenolic content of *Nerium oleander* extract, it is depicted that the value reached 281.35 mg GAE)/g dry extract.

Secondly, the water-soluble extract of *Nerium oleander* was analyzed using GC-MS, which revealed the presence of 14 components (Appendix A). The overall area of peaks of the discovered components are 47.12%, and the chemical structure of the identified compounds’ prospects are listed in Appendix A: 5-octadecenal (12.04%), 1-tridecanol (5.92%), 17-pentatriacontene (5.70%), (cis)-2-nonadecene (5.0%), and 1-hexacosanol (3.15%) that are the most often identified chemicals, accounting for 31.81% peak areas [41,42].

### 3.2. Characterization of the Biosynthesized AgNPs

The first indication for the formation of AgNPs is the color change during the chemical reaction, which is owing to the excitation of the surface plasmon vibration in AgNPs. The plant extract has a white off color, which was turned to deep yellow color after the addition of silver nitrate and stirring for a period affirming the formation of AgNPs.

#### 3.2.1. UV–Vis Spectroscopy Study

The absorption spectra of AgNPs wer© investigated using UV–vis spectroscopy as a well-known method for determining the production and stability of AgNPs in an aqueous solution [43]. As shown in Figure 1a, the fabricated extract has no peaks in the scanned UV-vis wavelength range demonstrating that the further appearance peaks will be attributed to the formed AgNPs. Meanwhile, Figure 1a shows that the produced AgNPs have a single SPR band with a maximum wavelength at 404 nm, indicating that the silver ion has been reduced to metallic silver [44]. The absorption peak grows sharper with completing the reaction by the formation of a spherical and homogenous dispersion of AgNPs. Furthermore, this photoionization causes a delayed reduction process, and the resulting AgNPs have a spherical shape [45]. In addition, temperature-assisted aging increases the development and narrowing of these generated particles’ size distributions [46].

#### 3.2.2. XRD Analysis

As shown in Figure 1b, the XRD patterns of AgNPs indicate the development of the silver crystalline structure via the appearance of peaks at 38.17°, 44.34°, 64.72°, and 77.15°, which could be assigned to 111, 200, 220, and 311 crystalline structures of the face centered cubic (fcc) produced silver nanocrystal, respectively, according to the XRD pattern (JCPDS file. No: 087-0720) [47,48]. These peaks revealed that silver was the core part of nanoparticles, with no extraneous peaks indicating that there are no contaminants associated with AgNPs as observed from XRD patterns. Debye–equation Scherrer’s was used to calculate the average crystalline size of AgNPs. The particle’s average size was calculated to be 4.3 nm by measuring the width of (111) Bragg’s reflection. The sizes of AgNPs estimated by XRD assessments are in accordance with those obtained using TEM and DLS techniques.

#### 3.2.3. TEM

The formation of AgNPs from silver nitrate in the presence of plant extracts was verified using TEM examination. At two different magnifications, the particle shape of the produced AgNPs was examined, and the results are displayed in Figure 2a,b. Mostly, AgNPs produced via polymer reduction and stabilization result in particles distribution size less than 20 nm, which agrees with our findings [34]. The average distribution size of AgNPs from TEM images was calculated using Image J4s software and it was around 6 nm (Figure 2c). The “d” spacing was calculated to be 0.235 nm, which matches the “d” spacing of the face centered cubic structure of metallic silver in the (111) orientation (Figure 2c). The circular fringes corresponding to the (111), (200), (220), and (311) planes of the face centered cubic of Ag are depicted in from SAED pattern. As a result, the SAED pattern’s results (Figure 2d) are in good agreement with the XRD pattern (Figure 1b), implying the formation of polycrystalline AgNPs.

#### 3.2.4. Dynamic Light Scattering (DLS)

The average diameter of distributed AgNPs in the liquid was intended using dynamic light scattering (DLS). Figure 2e displays the distribution size characteristic of the biosynthesized AgNPs as determined by the DLS technique. The strong signal verifies the equi-size particle distribution, which is an indication of monodispersity and matches with the UV–vis spectrum nicely. With a polydispersity index of 0.034, the distribution size profiles showed one size of AgNPs with an average size of 2.014 nm. Actually, it was observed that there was no significant difference between the average size from TEM and DLS.

#### 3.2.5. Zeta Potential

The degree of apparent electric charge surrounding the surface of AgNPs was measured by zeta potential. The charge of a nanoparticle was measured by the ion concentration inside an opposing charge at the surface of AgNPs after it has a total surface charge. For stable nanosuspension, a minimum of ±30 mV zeta potential values are needed [49]. The zeta potential of the biosynthesized AgNPs was determined to be −31 mV (Figure 2f), which corresponds to their excellent stability. These results are reliable with those obtained by Netala et al., [50]. Biosynthesized AgNPs induced by plant extract has a high negative zeta potential and are hence resistant to agglomeration. The negative number verifies particle repulsion and, the stability of the formulation

### 3.3. Characterization of AgNPs Loaded Nanofibers (AgNPs@NFs)

#### 3.3.1. SEM, EDX, Roughness and Contact Angle of AgNPs@NFs

Three different concentrations of AgNPs were used to be incorporated into electrospun nanofibers based on HPS/PU blend as depicted in Table 1. The produced electrospun nanofibers were coded as AgNPs-0@NFs, AgNPs-1@NFs, AgNPs-2@NFs and AgNPs-3@NFs.The electrospinning considerations were optimized and the morphology of the formed fibers was analyzed with the SEM technique. The images of SEM were taken at two different magnifications (2500× and 7500×). On the other hand, all SEM images of HPS/PU nanofibers unloaded and loaded with different concentrations of AgNPs are uniformly prepared with no beads or heterogeneous fibers, uniform and smooth nanofibers (Figure 3). The formation of beads free nanofibers was occurred due to the high spinnability of PU and the low concentration of the utilized HPS.

It is remarkable that the addition of AgNPs has no impact on the morphology of nanofibers in terms of short fibers and beads fibers and has no influence on the formation of continuous jet fibers. Moreover, it was observed that most of the small spherical sizes of AgNPs were incorporated into the porous structure of the prepared nanofibers.

Additionally, based on the SEM image, the diameter of nanofibers was gradually increased with increasing the concentration of AgNPs and the pore sizes of the electrospun sheet are in the range of 75 to 350 nm, which were changed with the fiber diameter.

To affirm the presence of AgNPs incorporated nanofibers, the sample of nanofibers coded with AgNPs-3@NFs was selected for further characterization using elemental analysis X-ray analysis (EDX). The element analysis, weight, and atomic percent of each element comprising the nanofibers were displayed in Figure 4a, b. In addition, the distribution of each element (C, O, Ag and N) is also displayed in Figure 4. As shown in (Figure 4), there are many elements in the analyzed sample. The existence of C and O can be attributed to the existence of HPS and PU. Meanwhile, the N atom is ascribed to PU. The presence of the Ag element is ascribed to the presence of AgNPs incorporated into the nanofibers.

#### 3.3.2. Surface Roughness of AgNPs@NFs

After studying the morphological features of the nanofibrous mats that loaded with different concentrations of AgNPs, The selected samples of SEM images were evaluated via Gwyddion 2.45 software to clarify the surface roughness of the prepared nanofibrous mats. As shown in Figure 5a–d and the reported data in Table 2, the roughness of HPS/PU (AgNPs-0@NFs) seems to be increased upon the addition of AgNPs. The roughness average (*R_a_*) was increased from 0.1061 to 0.1271 nm, however, the root means square roughness (*R_q_*) increased from 0.1353 to 0.1557 nm for the lowest and highest concentration of AgNPs, respectively. The maximum roughness height (*R_t_*) started from 0.5123 nm (AgNPs-0@NFs) to 0.7353 (AgNPs-1@NFs), 0.8628 (AgNPs-2@NFs), and 0.7865 nm (AgNPs-3@NFs). On the other hand, the maximum roughness valley depth (*R_v_*) increased from 90.2327 nm for AgNPs-0@NFs to 0.4149 nm for AgNPs-3@NFs with a high concentration of AgNPs. Furthermore, the relatively high values of *R_t_* vs. *R_v_* were observed significantly. The influence of *R_v_* values recompenses the irregular progression of *R_t_* values due to the combinations of *R_v_* and *R_t_*, which seems to be signified in *R_a_* values. The morphology of the surface is mainly affected by the crystal staking and the observed distortion typically causes a serial effect, particularly if it was commenced by a linear or surface defect type. These defects have higher energy than non-defected sites that promote the trap of ions via these energetic positions. This type of interaction incites the linkage between the surroundings through the chemical procedure.

#### 3.3.3. Contact Angle Measurements of AgNPs@NFs

The contact angle between the nanofibers sheet and the close water drops provides a simple idea around its ability to be coherent with the physiological milieu. As shown in Figure 6, the contact angle decreased from 147 ± 1.3 to 128.3 ± 0.5, 117.3 ± 1.1, 86.4 ± 1.6 and 55.7 ± 2.3°, for PU, AgNPs-0@NFs, AgNPs-1@NFs, AgNPs-2@NFs, and AgNPs-3@NFs, respectively due to the high ability of HPS to boost the surface defects over the surface of PU that it is vital for the contact with surroundings. This observation in addition to the wettability of the nanofibrous scaffolds is important to evaluate the interaction between the scaffolds and the ambient atmosphere. In addition, the surface roughness of the nanofibers along with their high porosity has been contributed to the hydrophobic characteristics of PU nanofibers. Furthermore, the addition of HPS to PU increases hydrophilicity. The implant fixation after surgical prosthesis depends mainly on the ability of the new materials to initiate the chemical interaction, as well as the physical interlocking within the ambient media [21].

#### 3.3.4. FTIR/ATR Spectra and TGA of AgNPs@NFs

FTIR spectroscopy was used to investigate the interaction between HPS, PU, and AgNPs. FTIR is well-known as an effective instrument for interpreting structural data. The FTIR spectra of HPC, PU, and HPS/PU nanofibers loaded with AgNPs (AgNPs@NFs) in the wavenumber range of 4000 to 400 cm^−1^ are shown in Figure 7. According to the HPS spectrum (Figure 7a), the broadness of the OH band detected at 3398 cm^−1^ related to the polymeric association of the hydroxyl group, while the peaks appeared at 2876 cm^−1^ correlated to the C-H stretching vibration band. There is also a peak at 1649 cm^−1^, indicating that the O-H band is observable. As displayed in Figure 7b, the C-O stretching band is reliable for the assigned peak at 1054 cm^−1^. The absorption band at 3327 cm^−1^ in the FTIR spectra of PU correlates to NH stretching. The strong two peaks at 2845 cm^−1^ and 2937 cm^−1^ are connected with CH_2_ stretching, whilst bands from 1478 cm^−1^ to 1507 cm^−1^ identify different types of -CH_2_ vibrations. The absorption band at 1734 cm^−1^ is also associated with a C=O group. The bands at 1535 cm^−1^ identify the group of NH vibrations. Hydrogen bonding between N-H and C=O groups is assigned at 1735 cm^−1^. Nonhydrogen-bonded carbonyl groups are represented by the band at 1721 cm^−1^ [51].

The stretching of the –NH and –OH groups in the spectrum of AgNPs@NFs gets wider as well as switches to 3336 cm^−1^, according to the correlating spectra of nanofibrous composite nanofibrous mats (Figure 7c). The existence of hydrogen-bond structures in these mixes causes the peak shift. Owing to variation in inter- and intra-molecular interactions, the other characteristic vibration bands from HPS overlap with those of PU, but the corresponding peaks are altered to lower wavenumbers [52]. The HPC was successfully incorporated into the PU nanofibrous mats, according to these findings. In addition to the foregoing band assignments, the presence of AgNPs is confirmed by the band at about 588 cm^−1^ [53,54]. The surface of AgNPs is likely covered with organic species generated from plant extracts, according to the FTIR spectroscopic finding.

Thermal gravimetric analysis (TGA) was used to outline the thermal stability of the nanofibers sheet. AgNPs-0@NFs and AgNPs-3@NFs were selected for such issue. The weighted samples were heated from room temperature to 500 °C at a rate of 10 °C/min. From Figure 8, it is clearly seen that there are three phases or steps for the degradation of nanofibers. The first zone is located at 50–95 °C, which is attributed to the evaporation of residual moisture and physically absorbed water. The two selected samples have the same temperature for the evaporation of water. The second degradation step, which is located at 250–270 °C is ascribed to the evaporation of polymer formed nanofibers (PU and HPS). Additionally, when the nanofibers are loaded with AgNPs, the degradation of polymer chains (PU and HPS) occurred at a high temperature of 264 °C when compared with that of nanofibers in absence of AgNPs (252 °C). Upon, increasing the temperature to 500 °C, about 18% and 22% of the weighted samples; AgNPs-0@NFs and AgNPs-3@NFs are still with no degradation with observing that the sample contains AgNPs has the high content. This indicated that the use of Ag as a nanofiller improved the polymer’s excellent heat stability.

### 3.4. Evaluation of the Antimicrobial Activity of AgNPs@NFs

#### 3.4.1. Antimicrobial Activity Assessment

The biosynthesis of AgNPs has attracted the academic society’s consideration owing to its multiple applications in biological control, sensing, catalysis, and pharmaceuticals [55]. The beneficial and cost-effective green synthesis of NPs using medicinal plant extracts has much more benefits [56]. Moreover, biogenic manufactured AgNPs could be used as an alternative to market antibiotics’ in the treatment of human-associated pathogenic microbes. The antibacterial effectiveness of biologically designed AgNPs against pathogenic microbes has been considerably recognized [57].

The potential antimicrobial action of the fabricated nanofibers (AgNPs-0@NFs, AgNPs-1@NFs, AgNPs-2@NFs, and AgNPs-3@NFs) towards tested pathogens microbes, including *P. aeruginosa*, *E. faecalis, C. albicans,* and *A. niger* was expressed as a diameter of ZOI (Table 3). The obtained results All nanofibers loaded with biogenic AgNPs revealed prominent antimicrobial potential towards the tested pathogenic species and the unloaded nanofibers showed no potential antimicrobial effect and did not display any ZOI. The results indicated that AgNPs-3@NFs showed the highest antimicrobial activity with ZOI diameters of 26 ± 0.23, 24 ± 0.25, 23 ± 0.23, and 21 ± 0.17 mm, against *P. aeruginosa*, *E. faecalis, C. albicans,* and *A. niger*, respectively. On the other side, AgNPs-1@NFs exhibited the lowest antimicrobial activity with ZOI diameters of 15 ± 0.20, 13 ± 0.18, 10 ± 0.16, and 11 ± 0.23 mm for *P. aeruginosa*, *E. faecalis, C. albicans,* and *A. niger*, respectively. In addition, the antimicrobial activity of the selected commercial antibiotic drugs showed low activity compared to AgNPs-1@NFs.

Moreover, the obtained results unveiled that the most susceptible tested microbes to all studied nanofibers was *P. aeruginosa*. These findings imply that the biosynthesized AgNPs can manage drug-resistant harmful microbes and could be exploited in the healthcare field. The research results on the antimicrobial property of biosynthesized AgNPs were in line with those of a prior investigation [58,59].

The estimated MIC values of fabricated nanofibers loaded with biogenic AgNPs against tested pathogens are graphically illustrated in Figure 9, Figure 10 and Figure 11. As shown in Figure 9, AgNPs-2@NFs could not completely inhibit the tested microbes. At the same time, the estimated MICs values of AgNPs-2@NFs against *P. aeruginosa* and *E. faecalis* were 250 and 500 mg/L within 30 min of exposure, respectively, while fungal species were more resistant than bacterial species (Figure 10). As shown in Figure 11, AgNPs-3@NFs strongly suppressed the proliferation of all tested species. The MIC values were 250 and 500 mg/L within 15 min for *P. aeruginosa* and *E. faecalis*, while *C. albicans* and *A. niger* required a longer time (30 min) to be destroyed with the same dose as bacterial species completely. This result demonstrated that the MICs of nanofibers against Gram-negative species were significantly less than the other tested microbes. In addition, previous experiments exhibited that Gram-negative bacteria’s better sensitivity to AgNPs than Gram-positive bacteria. This phenomenon is attributed to the differences in the biochemical composition between Gram-negative and Gram-positive bacteria [60].

The antibacterial activity of the biosynthesized AgNPs in our investigation is compatible with reported results [60,61]. Biosynthesized AgNPs had outstanding antibacterial performance towards *E. coli, K. pneumonia, P. aeruginosa, E. faecalis, Bacillus pumilus,* and *S. aureus*. The obtained results showed that the most excellent antibacterial activity was against Gram-negative strains, especially against *P. aeruginosa*, which is consistent with Kumar et al. [62]. A previous study [49] showed that the MIC of biosynthesized AgNPs is consistent with the antimicrobial property determined using disk diffusion. The smallest MIC was 6.25 g/mL for Gram-positive bacteria, which also had the maximum zone of inhibition. The lowest MIC values were recorded for *P. aeruginosa* (3.1 g/mL); the MIC data displayed that the biosynthesized AgNPs could inhibit the growth of bacteria at low doses.

Moving to the physiological variations for the investigated nanofibers, as shown in Figure 12, after contact with the new combined efficient dose (500 mg/L) of all examined NFs loaded with AgNPs, the population growth of all the tested microbial pathogens decreased significantly and considerably. The findings revealed that the slope in the bacterial growth curve was quicker and more prominent for *P. aeruginosa* bacteria, and the reduction rate was lesser for *C. albicans* microbes.

The findings in Figure 13 revealed that the amount of protein liberated increased substantially after subjecting damaged bacterial cells to a high concentration of AgNPs-1@NFs, AgNPs-2@NFs, and AgNPs-3@NFs. Protein leakage was more elevated in *P. aeruginosa* as a sign of Gram-negative bacteria, and it was more substantial than other species (Figure 13). On the other hand, the results displayed that the quantities of released protein from fungal strains were lower than the others, implying that these microbes were much more resistant. Due to the small cell wall, porous interstitial structures, and the formation of weak lipopolysaccharides, these results were supported by those obtained by Jiang et al., [63]. They reported that the rate of released protein and the quantities from the cells of *E. coli* were quicker and more significant than that observed for *S. aureus*. Therefore, there are expected morphological variations in the microbial cell wall and quantifiable cell material outflow upon applying the nanocomposite [64]. *E. coli* and *P. aeruginosa* as Gram-negative bacteria have a low ability to hold environmental stress. They resulted in loss or deformation owing to a lack of bacterial arrangements. Sensitively porous bacterial cells are produced by antimicrobial compounds [65].

Regarding to the kinetic modeling utilizing the pseudo-first-order kinetic model, Figure 14 revealed the inhibitory activity kinetics investigations of *P. aeruginosa*, *E. faecalis*, *C. albicans*, and *A. niger* in the presence of the effective concentrations AgNPs-3@NFs. The results showed that AgNPs-3@NFs, which could quickly suppress the growth of all tested pathogenic microbes, had the maximum inhibition for all evaluated human pathogens. Additionally, the findings demonstrated that *P. aeruginosa* suppressed proliferation quicker than other bacteria; however, *A. niger* organisms had the lowest inhibition frequency. It is worth noting that the tested fungal species were persistent over a long period. The findings demonstrated that in *P. aeruginosa* > *E. faecalis* > *C. albicans* > *A. niger*, the rate of inactivation of AgNPs-3@NFs as a *K_1_* constant was quick (Table 4).

#### 3.4.2. Mode of Action

The antibacterial action of AgNPs was associated with numerous different mechanisms. The globally acknowledged mechanism was that positively charged silver ions in AgNPs can easily engage with negatively charged sulfur or phosphorus-containing bio-macromolecules, such as nucleic acids and proteins, causing structural alterations in microbial species.

Gram-positive bacteria have a thick layer of peptidoglycan in their cell walls; this helps them to be more resistant to antimicrobial agents. While, Gram-negative bacteria contain a trace amount of polymers, which cannot protect them from the effects of bactericides. A detailed explanation of the mechanisms may well describe why and how to study AgNPs@NFs loaded with their biogenic AgNPs more solid inhibitory effect against Gram-negative bacteria than Gram-positive bacteria. The cell membrane contains fatty polysaccharides and phospholipids. Because of this dynamic diversification, their cell membranes have varying negative charges. Apart from that, the cell wall of Gram-negative bacteria has a higher negative charge than that of Gram-positive bacteria, and the Gram-positive cell wall has a more robust and deeper peptidoglycan layer than the Gram-negative cell wall. Due to the obvious differences in the properties of the two types of bacteria, Gram-negative bacteria had better penetration and electrostatic interaction with NPs than Gram-positive bacteria, resulting in Gram-negative bacteria having a greater inhibitory effect than Gram-positive bacteria [66].

### 3.5. Toxicological Performance Assay

The toxicity levels of the studied nanofibers were quantified using the MicroTox analyzer 500 for cytocompatibility assessments. The biocompatibility of the examined nanofibers was measured and summarized in Table 4. In vitro toxicity measurements of the investigated nanofibers were undertaken against the bioluminescent marine bacterium ‘*Aliivibrio fischeri*’. Actual EC_50%_ values for all tested nanocomposites AgNPs-0@NFs, AgNPs-1@NFs, AgNPs-2@NFs, and AgNPs-3@NFs after 15 min were 286, 268, 265, and 251%, respectively, according to the results.

The results revealed that all of the EC_50_ % values were greater than 100% at various intervals, implying that all tested nanofibers were friendly and non-toxic (Table 4). These results are in excellent agreement with the reported results by researchers who investigated the toxicity of nanoparticles against human breast adenocarcinoma (MCF7) and Hep2 cells and marine bacterium *‘Aliivibrio fischeri*’ [67].

## 4. Conclusions

Hydroxypropyl starch (HPS) as a modified natural polymer can be blended with a synthetic polymer, such as polyurethane (PU) to produce fabricated nanofibers (NFs) that can act as efficient wound dressing sheet after incorporation with biosynthesized AgNPs. AgNPs were prepared easily using a plant extract of *Nerium oleander* leaves that is abundant in desert roads and requires no effort in its cultivation. AgNPs were prepared in monodispersed form with a diameter less than 6 nm with good stability (−31 mv). The hydrophilicity of the fabricated nanofibers increased upon the addition of HPS to PU, which help in the initiation of the chemical interaction, as well as the physical interlocking within the ambient media of wounds. Different concentrations of AgNPs were incorporated into HSP/PU nanofibers to yield nanofibers sheets (AgNPs-1@NFs, AgNPs-2@NFs and AgNPs-3@NFs). The results proved that the nanofibers loaded with AgNPs has uniform fibers and the fibers’ diameter was increased with increasing the concentration of AgNPs. AgNPs-3@NFs loaded with biosynthesized AgNPs exhibited outstanding antimicrobial properties against human-associated pathogenic strains, including *P. aeruginosa, E. faecalis, C. albicans,* and *A. niger*. The physiological changes in microbial growth manners were quite different amongst the tested pathogens; this investigation could help understand the antibacterial mechanism of biosynthesized AgNPs for both Gram-positive and Gram-negative pathogenic bacteria and encourage the utilization of AgNPs as antimicrobial agents in biological applications. Furthermore, the toxicological screening results revealed that the nanofibers loaded with AgNPs were safe and biocompatible for biomedical applications.

## Data Availability

The data presented in this study are available on request from the corresponding author.

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
