# Peer review of "Fabrication of Nanofibers Based on Hydroxypropyl Starch/Polyurethane Loaded with the Biosynthesized Silver Nanoparticles for the Treatment of Pathogenic Microbes in Wounds"

_polymers, 2022, doi:10.3390/polym14020318_

Round 1
Reviewer 1 Report
In the manuscript “Fabrication of nanofibers based on hydroxypropyl starch/poly-2 urethane loaded with the biosynthesized silver nanoparticles for the treatment of pathogenic microbes in wounds”, the author reported the fabrication of environmentally wound dressing of electrospun nanofibers based on blending the modified natural polymer and synthetic biopolymers with the biosynthesized AgNPs.. The paper fit the aims and scope of Polymers. In general the paper makes fair impression and my recommendation is that it merits publication in this Journal’ However, there are some issues, which should be addressed before it can be accepted.
Line 21:Uv-vis, TEM, DLS and XRD should be defined.
In my opinion, it may be better to introduce the significance of research from the perspective of sustainable development. This work is beneficial to promote deep processing and improve economic benefits, which can increase environmental benefits and better highlight sustainability. It is consistent with the idea of eco-innovation, which focused on the demand for sustainable development. The author should comment on studies on the utilization of high-value target substances from by-products or wastes. Therefore, the incipit has to be supported with proper suitable literatures. Some very recent references on the reuse of waste or by-products, which achieving sustainable development in another approach, should be mentioned.
doi: 10.1016/j.lwt.2021.111617, doi:10.3390/polym13132044
Besides metal nanoparticles, other antibacterial nanomaterials, especially nanofibers, should be mentioned.
doi:10.3390/foods9040449
doi:10.1016/j.lwt.2020.109213
Preparation of antibacterial nanofibers by electrospinning should be reviewed.
doi:10.1021/acs.jafc.1c01351, doi:10.1016/j.indcrop.2021.114300,
Line 57: sheep
it is strongly suggested to indicate at the end of the Introduction section the main employed characterisation techniques in order to achieve their purpose.
Section 2.4 should be described with more detail.
Scheme 1 should be prepared in high resolution
Figure 1a: UV-vis spectroscopy of silver ion solution and the extra should be provided.
The statistical analyses should be described.
significant difference should be mentioned in Table 4
Author Response
Reviewer #1
In the manuscript “Fabrication of nanofibers based on hydroxypropyl starch/poly-2 urethane loaded with the biosynthesized silver nanoparticles for the treatment of pathogenic microbes in wounds”, the author reported the fabrication of environmentally wound dressing of electrospun nanofibers based on blending the modified natural polymer and synthetic biopolymers with the biosynthesized AgNPs.. The paper fit the aims and scope of Polymers. In general the paper makes fair impression and my recommendation is that it merits publication in this Journal’ However, there are some issues, which should be addressed before it can be accepted.
Line 21:Uv-vis, TEM, DLS and XRD should be defined.
- Thanks for your comment, all abbreviations have been identified.
In my opinion, it may be better to introduce the significance of research from the perspective of sustainable development. This work is beneficial to promote deep processing and improve economic benefits, which can increase environmental benefits and better highlight sustainability. It is consistent with the idea of eco-innovation, which focused on the demand for sustainable development. The author should comment on studies on the utilization of high-value target substances from by-products or wastes. Therefore, the incipit has to be supported with proper suitable literatures. Some very recent references on the reuse of waste or by-products, which achieving sustainable development in another approach, should be mentioned.
doi: 10.1016/j.lwt.2021.111617, doi:10.3390/polym13132044
Response:
Thanks a lot for your suggestions, the manuscript was updated with the suggested references, please see the revised manuscript.
Besides metal nanoparticles, other antibacterial nanomaterials, especially nanofibers, should be mentioned.
doi:10.3390/foods9040449, doi:10.1016/j.lwt.2020.109213
Response:
Thanks a lot for your suggestions, the suggested references have been added, please see the revised manuscript.
Preparation of antibacterial nanofibers by electrospinning should be reviewed.
doi:10.1021/acs.jafc.1c01351, doi:10.1016/j.indcrop.2021.114300,
Response:
Thanks a lot for your suggestions, the suggested references have been added, please see the revised manuscript.
Line 57: sheep
Response:
The word has been corrected to “cheap”
it is strongly suggested to indicate at the end of the Introduction section the main employed characterisation techniques in order to achieve their purpose.
Response:
Thanks for your comment. The characterisation techniques have been added at the end of introduction part. Please see the revised manuscript.
Section 2.4 should be described with more detail.
Response:
The characterization section (Section 2.4) has been revised and modified. Please see the revised manuscript.
Scheme 1 should be prepared in high resolution
Response:
Thanks for your observation. The scheme has been redrawn with high resolution
Figure 1a: UV-vis spectroscopy of silver ion solution and the extract should be provided.
Response:
Thanks for your useful comment, The UV-vis of the fabricated extract has been added. Meanwhile, UV-vis of AgNPs ions have been previously added in many published articles (absorbance at wavelength 340-380 nm). In addition, as observed from XRD data, there is no peaks for Ag ions (peak at 31o ) demonstrating the formation of nanoparticles free from impurities and all ions have been chemically converted to metallic Ag nanoparticles.
The statistical analyses should be described.
Response:
The statistical analyses has been added (2.6. Statistical analyses) as follow: “GraphPad Prism version 5.0 (USA) was applied to conduct the statistical study. The level of significance (a probability of p <0.05) between the doses of tested nanomaterial and viable cells of targeted microbes was calculated applying two-way analysis (ANOVA) of variance.”
significant difference should be mentioned in Table 4
Response:
Thanks for your comment, the significant difference has been added in Table 4. Each gained value is extracted from three replicates and is expressed as the mean with SEM in the same row; a b.c letters refer to significant differences between means (p < 0.05).
Reviewer 2 Report
Dear authors,
This paper deals with Fabrication of nanofibers based on hydroxypropyl starch/polyurethane loaded with the biosynthesized silver nanoparticles for the treatment of pathogenic microbes in wounds.
It contains interesting results, but some issues must be addressed to before being considered for publication.
Abtract section.
More quantitative data about your results must be included in the abstract.
Introduction section
Find a better place in the sentence to add the References 4-6 , maybe together with ref 7.
Line 45…the word energetic is not proper in this context, replace it.
Line 59..mat?...please define it.
Line 78… an
Material section
Line 88….80k g/ mol….. you can choose between 80.000 g/mol or in kDa, but not 80k g/mol
Line 98…being
Line 117…mats?
Line 118, eliminate the – and add a comma.
Line 122. Each characterization technique must be described in different rows and more details for each one must be add in this section.
Statistical analysis section must be included.
Results section.
Figure 1 quality must be improved, providing a higher resolution material.
Line 208---(AgNPs)-…remove the -
Line 218 to 221. It is not clear how you can find all those non-polar chemicals in a water extract ?. Moreover, GC-MS analysis is for volatile compounds isolated mainly in organic solvents (ethanol, acetone, cloroform, ethyl acetate) but not in water. For water extracts, HPLC analysis is prefered. You must clarify this point.
There is a lack of characterization techniques for the biocomposite electrospun fibers. It is strongly suggested to add some results from FTIR, TGA or XRD. Otherwise, it would be impossible to confirm the composition of electrospun nanofibers composite.
Line 226---at the end of the row remove -.
Line 232-233…how can you know they are spherical and homogeneous by UV visible?
Line 236… why ae you talking about particle size ditribution here? It would help if you can discuss what is happening at this wavelength, how different is regarding the one from other papers.Why it happens, etc.
Line 244 to 246..you are mixing concepts.. crystallinity and average size…and crystalliity data is not available. Please provide the equations used for calculation in Material section.
Line 258 to 260… It is not clear how do you determine d spacing…provide a higher resolution image of figure indicating the d spacing…and XRD plane for this EDS picture.
The value of obtained particle size is quite unusual for biosynthesized AgNPs using aqueous extract. Can you compare it with previous work and suggest how it was possible?. Considering the NPs rippeng effect, etc.
Figure 4. please provide the normal SEM picture to know the aspect of the measured fiber.
Figure 5. It looks as AFM analysis and not FESEM one…can you clarify this point, please. The measured data from Table 3 confirm this point.
Author Response
Reviewer #2
Dear authors,
This paper deals with Fabrication of nanofibers based on hydroxypropyl starch/polyurethane loaded with the biosynthesized silver nanoparticles for the treatment of pathogenic microbes in wounds.
It contains interesting results, but some issues must be addressed to before being considered for publication.
Abtract section.
More quantitative data about your results must be included in the abstract.
Response:
Thanks for your recommendation. The quantitative data has been added to the abstract part.
Introduction section
Find a better place in the sentence to add the References 4-6 , maybe together with ref 7.
Response:
Thanks for your recommendation. The required references have been combined.
Line 45…the word energetic is not proper in this context, replace it.
Response:
The word has been replaced.
Line 59..mat?...please define it.
Response:
The mat is defined as sheet. Thus. For clarification, the word has been replaced with more clear word (sheet).
Line 78… an
Response:
The sentence has been corrected with no strikethrough
Material section
Line 88….80k g/ mol….. you can choose between 80.000 g/mol or in kDa, but not 80k g/mol
Line 98…being
Response:
The sentence has been corrected with no strikethrough
Line 117…mats?
Response:
The word “mats” has been changed to “sheets”.
Line 118, eliminate the – and add a comma.
Response:
The mark (-) has been deleted and comma was added.
Line 122. Each characterization technique must be described in different rows and more details for each one must be add in this section.
Response:
Thanks for your comment. The details for each utilized technique have been added.
Statistical analysis section must be included.
Response:
The statistical analyses has been added (2.6. Statistical analyses) as follow: “GraphPad Prism version 5.0 (USA) was applied to conduct the statistical study. The level of significance (a probability of p <0.05) between the doses of tested nanomaterial and viable cells of targeted microbes was calculated applying two-way analysis (ANOVA) of variance.”
Thus, the significant difference has been added in Table 4. Each gained value is extracted from three replicates and is expressed as the mean with SEM in the same row; a b.c letters refer to significant differences between means (p < 0.05).
Results section.
Figure 1 quality must be improved, providing a higher resolution material.
Response:
Scheme 1 and figure 1 have been redrawn to be more readable.
Line 208---(AgNPs)-…remove the –
Response:
The sentence has been checked and corrected.
Line 218 to 221. It is not clear how you can find all those non-polar chemicals in a water extract ?. Moreover, GC-MS analysis is for volatile compounds isolated mainly in organic solvents (ethanol, acetone, cloroform, ethyl acetate) but not in water. For water extracts, HPLC analysis is prefered. You must clarify this point.
Response:
Nonpolar compounds such as essential oils were normally extracted by hot water and analyzed by GC mass and also low polar compounds could be detected by GC. Our target in this research is to measure the role of essential oils and low polar compounds in the biosynthesis of nanoparticles. In general, it was known that all plants contain phenolic compounds and their role in the biosynthesis of nanoparticles are previously hypothesized.
- https://www.researchgate.net/publication/334370205_Metal_Nanoparticles_Synthesis_through_Natural_Phenolic_Acids
- https://www.researchgate.net/publication/267453921_Optimization_of_pressurized_hot_water_extraction_of_Lavandin_essential_oils_via_central_composite_design.
There is a lack of characterization techniques for the biocomposite electrospun fibers. It is strongly suggested to add some results from FTIR, TGA or XRD. Otherwise, it would be impossible to confirm the composition of electrospun nanofibers composite.
Response:
Regarding the preparation of electrospun nanofibers, we aimed to prepare an environmentally nanofibers from PU and HPS. For electrospinning process, the mixture solution was prepared from the physical mixing of HPS and PU compounds. In addition, for nanofibers preparation containing AgNPs, the as prepared AgNPs was added with different concentrations. Thus, from our point of view, there is no need for FTIR or XRD tools. However, we completely agree with the reviewer comment about evaluating the thermal stability of the resultant nanofibers. Thus, we carried out the TGA for two selected samples; AgNPs-0@NFs and AgNPs-3@NFs. And the obtained data are set below:
“Thermal gravimetric analysis (TGA) was used to outline the thermal stability of the nanofibers sheet. AgNPs-0@NFs and AgNPs-3@NFs were selected for such issue. The weighted samples were heated from room temperature to 500°C at a rate of 10°C/min. From Figure 7, it is clearly seen that there are three phases or steps for the degradation of nanofibers. The first zone is located at 50- 95 °C, which attributed to the evaporation of water and the elimination of moisture. The two selected samples have the same temperature for the evaporation of water. the second degradation step, which is located at 250-270°C is ascribed to the evaporation of polymer formed nanofibers (PU and HPS). Additionally, when the nanofibers are loaded with AgNPs, the degradation of polymer chains (PU and HPS) occurred at high temperature 264 °C when compared with that of nanofibers in the absence of AgNPs (252 °C). Upon increasing the temperature to 500 °C, about 18% and22 % of the weighted samples; AgNPs-0@NFs and AgNPs-3@NFs are still with no degradation with observing that the sample contains AgNPs has the high content. This indicated that the use of Ag as a nanofiller improved the polymer's excellent heat stability.”
Line 226---at the end of the row remove -.
Response:
Thanks for your observation. The mark – has been removed.
Line 232-233…how can you know they are spherical and homogeneous by UV visible?
Response:
As stated in literature, the shape of the prepared AgNPs (spherical, cubic, rectangle, triangle and so on) can be depicted from UV-vis spectra as a first noticeable. Each form has a rang in the UV-vis. For example, the spherical shape of AgNPs can be depicted when its UV-vis exhibited absorbance in the range 400-450 nm. In addition, it can be clarified the shape of AgNPs form the formed color (yellow, dark yellow, green and so on.) Taking in mind that, it is just noticeable and it must be confirmed by other tools such as TEM.
The homogeneity can be illustrated from the shape of the peak. The sharp peak is an indication for the small range. From Figure 1b, that the peak of AgNPs has a clear absorbance mainly ranged from 400 - 430 with maximum absorbance at 404 nm.
Line 236… why are you talking about particle size ditribution here? It would help if you can discuss what is happening at this wavelength, how different is regarding the one from other papers.Why it happens, etc.
Response:
Thanks for your comment. After the preparation of AgNPs, it is necessary to fully characterize the nanoparticles in terms of particle shape, DLS, zeta potential and size distribution. Thus in our work, for evaluating the nanoparticles distribution, we performed two analysis. The first one was focused on calculating the size distribution from TEM images via using software program (Image J4s) (Figure 2d). The accurate analysis for the size distribution was carried out using DLS technique. It was performed by measuring the average hydrodynamic size of AgNPs for 18 time and the resultant average size (2.014 nm) was obtained as graphical image (Figure 2e).Thus, it is logically to have different between the two different techniques. One of the them is focused on the evaluating in dry state (TEM and histogram) and the second characterization tool is DLS for measuring the average hydrodynamic size in its liquid state (DLS technique).
Line 244 to 246..you are mixing concepts.. crystallinity and average size…and crystalliity data is not available. Please provide the equations used for calculation in Material section.
Response:
XRD was used to determine the crystallinity of the prepared AgNPs. As shown in Figure 1(b), the XRD patterns of AgNPs indicate the development of the silver crystalline structure via the appearance of peaks at 38.17°, 44.34°, 64.72°, and 77.15°, which could be assigned to 111, 200, 220, and 311 crystalline structures of the face centered cubic (fcc) produced silver nano-crystal, respectively, according to the XRD pattern (JCPDS file. No: 087-0720) [47,48]. These peaks revealed that silver was the core part of nanoparticles, with no extraneous peaks indicating that there are no contaminants associated with AgNPs as observed from XRD patterns.
For measuring the size from XRD, Debye–equation Scherrer's ((D=0.9λ/β cos θ) to determine various characteristics of the crystalline material (Cullity and Stock, 2001), where D is the crystal size, λ is the wavelength of X-ray, θ is the Braggs angle in radians, and β is the full width at half maximum of the peak in radians.) was used to calculate the average crystalline size of AgNPs. The particle's average size was calculated to be 4.3 nm by measuring the width of (111) Bragg's reflection.
Additionally, in our work, we aimed to determine the average of AgNPs via three methods:
- Size distribution (Histogram) from TEM images.
- Average hydrodynamic size (DLS technique)
- XRD (through measuring the width of (111) Bragg's reflection
All these methods affirmed that the particle size of the prepared AgNPs is nearly 5 nm affirming the capability of extract to act as both reductant and stabilizing agent for forming well stability AgNPs with very small size.
The equations used for calculation the size of AgNPs has been also added to XRD section in materials and methods section.
Line 258 to 260… It is not clear how do you determine d spacing…provide a higher resolution image of figure indicating the d spacing…and XRD plane for this EDS picture.
Response:
Thanks for your comment, it was calculated from SAED image through TEM instrument. The high resolution of SAED image has been added with clear planes. As mentioned in our research work. the SAED pattern's results (Figure 2 c) are in a good agreement with the XRD pattern (Figure 1b)., implying the formation of polycrystalline AgNPs. The planes for SAED have been mentioned in the manuscript, which is in accordance with that of XRD.
The value of obtained particle size is quite unusual for biosynthesized AgNPs using aqueous extract. Can you compare it with previous work and suggest how it was possible?. Considering the NPs rippeng effect, etc.
Response:
- The size of the prepared AgNPs have been characterized via many tools; TEM, DLS and XRD. All these different tools affirmed that the nanoparticles are formed with very small size (around 5 nm).
- As stated in our manuscript that the plant extract comprises many compounds, aldehydes and alcohols, with reducing groups such as 5-octadecenal, 1-tridecanol, 1-hexacosanol and 1-docosanol, which can reduce silver ions (Ag+) to silver nanoparticles (AgNPs). The extract bearing also hydroxyl groups that can act as stabilizing agents for the produced AgNPs.
- Comparing with our obtained results, there many published papers such as the article https://www.hindawi.com/journals/ijmicro/2019/8642303/
entitled “Synthesis of Silver Nanoparticles Using Aqueous Extract of Medicinal Plants’ (Impatiens balsamina and Lantana camara) Fresh Leaves and Analysis of Antimicrobial Activity”. In this article, the authors aimed to synthesize Ag nanoparticles using aqueous extracts of fresh leaves of I. balsamina and L. camara and the data reveled that the nanoparticles have been prepared with very small size (3.2, 4, and 6, nm) using Lantana camara as plant extract. The obtained different size was mainly depended on the utilized concentration of silver nitrate (1 mM, 2 mM and 3 mM)
- In another published articles (Biosynthesis of Gold and Silver Nanoparticles Using Emblica Officinalis Fruit Extract, Their Phase Transfer and Transmetallation in an Organic Solution) and Green Synthesis of Silver Nanoparticles Involving Extract of Plants of Different Taxonomic Groups, the authors mentioned that AgNPs was prepared with very small size (5 nm) using Punica granatum as plant extract
https://medcraveonline.com/JNMR/green-synthesis-of-silver-nanoparticles-involving-extract-of-plants-of-different-taxonomic-groups.html
and
https://www.ingentaconnect.com/contentone/asp/jnn/2005/00000005/00000010/art00010
Figure 4. please provide the normal SEM picture to know the aspect of the measured fiber.
Response:
Thanks for your comment. However, the resultant EDX image and their elemental and mapping analysis were fabricated from Figure 3h. For clarification, the nanofibers samples were firstly scanned via SEM and after sectioning the suitable image, the technical person that responsible for measuring select very small area from SEM image to characterize it and to identify the elements that presented in this very small area.
In some cases, which have already been implemented publications, it is possible to use an image other than the one we used to characterize the morphological structure of the prepared sample. But in this work, Figure 3 (h) was selected to designate the elements analysis of the prepared nanofibers (AgNPs-3@NFs) using very small area of ​​the image.
Figure 5. It looks as AFM analysis and not FESEM one…can you clarify this point, please. The measured data from Table 3 confirm this point.
Response:
The data in Figure 5 was focused on evaluating the surface roughness of the nanofibers samples. As presented in the characterization section, the surface roughness of the micrographs produced by FESEM was studied using Gwyddion 2.45 software. Based on that, the image of SEM was taken to be evaluated via Gwyddion 2.45 software and the Thus, this data is nearly similar to AFM data.
Round 2
Reviewer 2 Report
Dear authors,
The manuscript was improved but still, some issues must be attended to before being considered for publication.
It is fully understood that your starting materials are two commercial polymers. But in this manuscript, no information that allows it to confirm this matter must be provided. That is why an FTIR/ATR analysis of your samples is requested.
In the case of the extract, the preparation methodology is still confusing (there are missing some steps). For essential oils (that is consistent with the composition that you provided) a simple water boiling and filtering of the extract are not enough to obtain in an appropriate form (please clarify). In another paragraph, you also mentioned the polyphenolic presence in extracts (but o information is available). By the way, essential oils are not mixible with water.
The methodology used for obtaining the FESEM images does not correspond to FESEM maybe to AFM, please check them again.
Author Response
Dear Editor:
Thank you for your letter and the opportunity to revise our manuscript entitled “Fabrication of nanofibers based on hydroxypropyl starch/polyurethane loaded with the biosynthesized silver nanoparticles for the treatment of pathogenic microbes in wounds”polymers-1543372. The suggestions offered by the reviewers have been immensely helpful. We have studied comments carefully and have made corrections highlighted wit yellow color in the revised manuscript. The main corrections and adjustments in the paper and the response to the reviewers’ comments are as follows
Comments and Suggestions for Authors
The manuscript was improved but still, some issues must be attended to before being considered for publication.
Response
- Thanks for your comment.
It is fully understood that your starting materials are two commercial polymers. But in this manuscript, no information that allows it to confirm this matter must be provided. That is why an FTIR/ATR analysis of your samples is requested.
Response
- Thanks for the reviewer recommendation FTIR/ATR has been performed for the sample powder of HPS, PU and nanofibrs of them and loaded with AgNPs as follow:
- FTIR spectroscopy was used to investigate the interaction between HPS, PU, and AgNPs. FTIR is well-known as an effective instrument for interpreting structural data. The FTIR spectra of HPC, PU, and nanofibers of both and loaded with AgNPs in the wavenumber range of 4000 to 400 cm-1 are shown in Figure 7. According to the HPS spectrum (Figure 7a), the broadness of OH band detected at 3398 cm-1 related to the polymeric association of the hydroxyl group, while the peaks appeared at 2876 cm-1 correlated to the C-H stretching vibration band. There is also a peak at 1649 cm-1, indicating that the O-H band is observable. As displayed in Figure 7 (b), the C-O stretching band is reliable for the assigned peak at 1054 cm-1. The absorption band at 3327 cm-1 in the FTIR spectra of PU correlates to NH stretching. The strong two peaks at 2845 cm-1 and 2937 cm-1 are connected with CH2 stretching, whilst bands from 1478 cm-1 to 1507 cm-1 identify different types of -CH2 The absorption band at 1734 cm-1 is also associated to a C=O group. The bands at 1535 cm-1 identify the group of NH vibrations. Hydrogen bonding between N-H and C=O groups are assigned at 1735 cm-1. Nonhydrogen-bonded carbonyl groups are represented by the band at 1721 cm-1 [1].
- The stretching of the –NH and –OH groups in the spectrum of HPS/PU@AgNPs gets wider as well as switches to 3336 cm-1, according to the correlating spectra of nanofibrous composite nanofibrous mats (Figure 7 c). The existence of hydrogen-bond structures in these mixes causes the peak shift. Owing to variation in inter- and intra-molecular interactions, the other characteristic vibration bands from HPS overlap with those of PU, but the corresponding peaks are altered to lower wavenumbers [2]. The HPC was successfully incorporated into the PU nanofibrous mats, according to these findings. In addition to the foregoing band assignments, the presence of AgNPs is confirmed by the band about 588 cm-1 [3,4]. The surface of AgNPs is likely covered with organic species generated from plant extracts, according to the FTIR spectroscopic finding.
References
- Asefnejad, A.; Khorasani, M.T.; Behnamghader, A.; Farsadzadeh, B.; Bonakdar, S. Manufacturing of biodegradable polyurethane scaffolds based on polycaprolactone using a phase separation method: physical properties and in vitro assay. Int. J. Nanomedicine 2011, 6, 2375.
- Mura, P. Analytical techniques for characterization of cyclodextrin complexes in aqueous solution: a review. J. Pharm. Biomed. Anal. 2014, 101, 238–250.
- Hussein, E.A.M.; Mohammad, A.A.-H.; Harraz, F.A.; Ahsan, M.F. Biologically synthesized silver nanoparticles for enhancing tetracycline activity against staphylococcus aureus and klebsiella pneumoniae. Brazilian Arch. Biol. Technol. 2019, 62.
- Tripathi, S.; Mehrotra, G.K.; Dutta, P.K. Chitosan–silver oxide nanocomposite film: Preparation and antimicrobial activity. Bull. Mater. Sci. 2011, 34, 29–35.
In the case of the extract, the preparation methodology is still confusing (there are missing some steps). For essential oils (that is consistent with the composition that you provided) a simple water boiling and filtering of the extract are not enough to obtain in an appropriate form (please clarify). In another paragraph, you also mentioned the polyphenolic presence in extracts (but o information is available). By the way, essential oils are not mixible with water.
Response
Thanks for the reviewer recommendation, extraction and biosynthesis of silver nanoparticles have been rewritten to be more readable as follow:
- Silver nanoparticles have been biosynthesized by the method previously described by Subbaiya et al., [5]with some modifications. Briefly, the fresh leaves of Nerium oleander (ornamental plant) were collected from plants grown in the streets of Mansoura city, Egypt during September 2021. For preparing the plant extract, Leaves were washed with tap water to remove dirties and dust followed wiping with filter paper to remove any residual water drops. The plant extract was prepared by mixing 20g of green leaves with 500 mL of boiling distilled water for 30min. the produced water extract was filter through Whatman No. 40 filter paper to get rid of any particulate materials. For the biosynthesis of silver nanoparticles, 25 mL of 2 mM aqueous solution of silver nitrate was added to 50 mL of the extract with continuous stirring at 25ºC for 24 h. During stirring, it was noted that the colorless solution was turned into brownish red color signifying the formation of AgNPs.
- The results indicated that the Nerium oleander extract exhibited total phenolic content of 281.35mg GAE)/ g dry extract.
- Subbaiya, R.; Shiyamala, M.; Revathi, K.; Pushpalatha, R.; Selvam, M.M. Biological synthesis of silver nanoparticles from Nerium oleander and its antibacterial and antioxidant property. Int. J. Curr. Microbiol. Appl. Sci 2014, 3, 83–87.
- In addition, the total phenolic content (as polar compounds) of Nerium oleander extract has been measured to estimate their role in the biosynthesis of silver nanoparticles. Moreover, we selected the GC/mass analysis to measure the nonpolar contents of the same extract. We did not prepare the essential oil but we just estimated that GC/mass to give us an indication about the presence of nonpolar content of the extract. Generally, our target was not to prepare phenolic compound or oils but to biosynthesize silver nanoparticles by plant extract and used the as prepared nanoparticles in nanofibers formation to act as efficient wound dressing nanofibers. So, we just added a discussion about the responsible compounds for the biosynthesis of nanoparticles.
- The total phenolic content of Nerium oleander extract was measured by Folin-Ciocalteu’s reagent method as previously described by Kumar et al., [36]. In details, the reaction mix consisted of 100 µL of plant extract, 200 µg/ml, 500 µL of the Folin-Ciocalteu's reagent and 1.5 mL of sodium carbonate (20%). The mixture was shaken and made up to 10 mL using distilled water. The mixture was allowed to stand for 2 h, and then the absorbance was measured at 765 nm. Gallic acid was used as standard. All determinations were carried out in triplicate. The total phenolic content was expressed as mg gallic acid equivalent (GAE) per g extract.
The methodology used for obtaining the FESEM images does not correspond to FESEM maybe to AFM, please check them again.
Response
- Thanks for your comment. As mentioned in the experimental part, For evaluating the surface roughness of the prepared nanofibrous mats, we used a program called Gwyddion 2.45 software (not instrument). This software produced figures similar to that of AFM. Therefore, we confirm that we used a software program and did not use AFM instrument. We mentioned that in the experimental apart as follow: The surface roughness of the prepared nanofibrous mats were studied using Gwyddion 2.45 software.
Round 3
Reviewer 2 Report
No comments